# High-Risk Screening for Fabry Disease: A Nationwide Study in Japan and Literature Review

**DOI:** 10.3390/diagnostics11101779

**Published:** 2021-09-27

**Authors:** Takaaki Sawada, Jun Kido, Keishin Sugawara, Kimitoshi Nakamura

**Affiliations:** 1Department of Pediatrics, Faculty of Life Sciences, Kumamoto University, 1-1-1 Honjo, Kumamoto City 860-8556, Japan; sawada.takaki@kuh.kumamoto-u.ac.jp (T.S.); sugawara.kei2552@gmail.com (K.S.); nakamura@kumamoto-u.ac.jp (K.N.); 2Department of Pediatrics, Graduate School of Medical Sciences, Kumamoto University, Kumamoto City 860-8556, Japan

**Keywords:** α-galactosidase A, Fabry disease, *GLA*, high-risk screening, newborn screening

## Abstract

Fabry disease (FD) is an X-linked inherited disorder caused by mutations in the *GLA* gene, which encodes the lysosomal enzyme α-galactosidase A (α-Gal A). FD detection in patients at an early stage is essential to achieve sufficient treatment effects, and high-risk screening may be effective. Here, we performed high-risk screening for FD in Japan and showed that peripheral neurological manifestations are important in young patients with FD. Moreover, we reviewed the literature on high-risk screening in patients with renal, cardiac, and central neurological manifestations. Based on the results of this study and review of research abroad, we believe that FD can be detected more effectively by targeting individuals based on age. In recent years, the methods for high-risk screening have been ameliorated, and high-risk screening studies using *GLA* next-generation sequencing have been conducted. Considering the cost-effectiveness of screening, *GLA* sequencing should be performed in individuals with reduced α-Gal A activity and females with certain FD manifestations and/or a family history of FD. The findings suggest that family analysis would likely detect FD patients, although *GLA* sequencing of asymptomatic family members requires adequate genetic counseling.

## 1. Introduction

Fabry disease (FD; MIM 301500) is an X-linked inherited disorder caused by mutations in the *GLA* gene and resulting in the impaired activity of the lysosomal enzyme α-galactosidase A (α-Gal A; EC 3.2.1.22) [1]. α-Gal A deficiency causes globotriaosylceramide (Gb3) accumulation in cells throughout the body, leading to various clinical manifestations. Moreover, globotriaosylsphingosine (Lyso-Gb3), which is the deacylated form of Gb3, is cytotoxic and exerts inflammation-inducing and fibrosis-promoting effects [2]. 

Based on the clinical manifestations and onset age, men with FD are classified into two types: the classical type, where patients develop symptoms, such as limb pain, acroparesthesia, sweating disorders, angiokeratoma, and/or gastrointestinal symptoms in childhood or adolescence; and the late-onset type, where patients develop kidney, heart, and cerebrovascular disorders in adulthood or at an older age. In addition, because FD is an X-linked inheritance, some females can develop these manifestations, and heterozygous female patients develop symptoms similar to those of the classical type, suggesting the involvement of lyonization [3].

Enzyme replacement therapy (ERT) is a currently available treatment option for FD that slows renal deterioration, alleviates the progression of cardiomyopathy, and prevents morbidity and death [4]. Moreover, a pharmacological chaperone therapy (Migalastat; 1-deoxygalactonojirimycin) was recently approved for FD patients with amenable *GLA* variants [5]. Migalastat that selectively and reversibly binds to the active sites of amenable mutant forms of α-Gal A enzyme [6,7]. Given its efficacy, convenient oral regimen and the limited therapeutic options available, migalastat can be an important therapeutic option for FD patients with migalastat-amenable *GLA* variants [8]. Responsiveness of each variant should be carefully evaluated [9,10,11]. Early treatment is critical to preserve organ function; however, many patients are diagnosed at later stages or misdiagnosed owing to the nonspecific clinical manifestations of the early stages of the disease [12,13]. Newborn screening (NBS), which is effective for early detection of patients with FD, revealed that FD is more frequent than expected in Japan and worldwide [14,15,16,17,18], suggesting that there might be many undiagnosed patients with FD. High-risk screening is one method of FD detection via uniform screening in patients with FD-related symptoms, including renal, cardiac, and central neurological manifestations. Various high-risk screening programs are performed worldwide using different target populations and screening methods. In Japan, we previously performed high-risk screening in patients with risk factors for FD [13], and gene and substrate-reducing therapies for FD are currently undergoing clinical trials worldwide [19]. The approval of these new treatments will make high-risk screening for FD increasingly important because patients with FD can be treated using these new medicines. Here, we review the outcomes of the high-risk screening performed in our study and those worldwide and discuss future methods of high-risk screening.

## 2. Targets for High-Risk Screening of FD

The clinical course of FD varies depending on the disease type. The classical type, which includes some heterozygous females, shows symptoms, such as limb pain, acroparesthesia, sweating disorders, and/or gastrointestinal symptoms in childhood, with cardiac, renal, and cerebrovascular diseases appearing in adulthood and older people [20]. However, in most cases, limb pain or acroparesthesia disappears in adulthood and older people [21]. The late-onset type develops cardiac, renal, and/or cerebrovascular diseases after adolescence and without classical manifestations [22]. In some NBS, late-onset variants are detected more often than classical variants [14,15]. Additionally, patients with late-onset FD might be more common than those with classical FD; therefore, high-risk screening is important for individuals with FD-related manifestations and who do not receive NBS. Most high-risk screening targets patients with cardiac, renal, or cerebrovascular diseases. Specifically, those presenting left ventricular hypertrophy (LVH) or hypertrophic cardiomyopathy (HCM) related to cardiac disease, undergoing dialysis and/or kidney transplants for renal disease, and younger patients with stroke-related cerebrovascular disease.

## 3. Methods for High-Risk Screening

There are several methods for FD screening, including measuring α-Gal A enzyme activity in the blood, Gb3 level in urine or blood and/or Lyso-Gb3 level in the blood, and *GLA* gene sequencing.

α-Gal A, which is extracted from dried blood spots (DBSs), plasma, or serum, reacts with artificial substrates, and its activity is measured by a fluorescence plate reader or tandem mass spectrometry [23]. DBS is useful for NBS because it is easy to collect, store, and transport and can be used to evaluate those of individuals simultaneously. However, it should be noted that decreases in α-Gal A activity might not be detectable in female patients with FD, even when using each assay method [24].

Moreover, it is possible that methods for measuring Gb3 accumulation in urine and blood [25,26] and Lyso-Gb3 in blood [27] might be insufficient for some female patients with FD [28]. Individuals are likely to have false-positive results when they harbor non-pathogenic polymorphisms. To prevent these false positives, high-risk screening using *GLA* sequencing should be performed. However, depending on the region of the *GLA* gene to be analyzed, there are some gene variants that cannot be detected. For example, deep intronic variants, such as IVS4+919G>A, are not detected during the analysis of exons and/or intron regions close to exons [29]. Conversely, analysis of the whole *GLA* gene might identify multiple variants of unknown significance (VOUS), including intronic variants, that are difficult to interpret. 

## 4. *GLA* Polymorphisms

Some variants result in amino acid substitutions and/or low enzyme activity but do not present FD-related manifestations. The p.E66Q, p.D313Y, p.A143T, p.S126G, and p.R118C variants are not regarded as pathogenic [30,31,32,33,34], and evaluation of these variants as pathogenic is controversial [35,36,37]. Therefore, we considered these five variants as polymorphisms without pathogenicity and subsequently calculated the frequency of patients with FD, excluding these five variants.

## 5. Algorithms for Screening and Diagnosis

For high-risk screening, individuals meeting the selection criteria receive information about FD and the form for informed consent. Possibly, screening tests might not have previously been undertaken in order to reduce the false-positive rate. In our high-risk screening algorithm, individuals showing low DBS α-Gal A activity (<a cut-off) in the two times tests (first and second assays) underwent *GLA* gene sequencing. Detection of a known pathogenic variant by *GLA* sequencing results in a definitive FD diagnosis; however, the detection of a novel variant or VOUS results in a diagnosis made comprehensively based on the clinical course, family history, α-Gal A activity level, Lyso-Gb3 level in the blood, and histological findings. Individuals with positive results receive a definitive diagnosis.

## 6. High-Risk Screening in Japan 

We had conducted NBS for FD in western Japan and reported that one in 7683 infants might develop FD [14]. This frequency is similar to that of NBS in the United States [18,38,39]. To investigate the effect of high-risk screening for individuals with FD-related manifestations not receiving NBS, we conducted high-risk screening using the NBS method for FD [13]. The outcome of the high-risk screening in Japan is summarized below.

We screened 18,135 individuals throughout Japan with at least one of the following five risk factors: renal manifestations (e.g., proteinuria, chronic kidney disease, diabetic nephropathy, mulberries in urine, and/or receiving dialysis), cardiac manifestations (e.g., LVH on electrocardiography or echocardiography), central neurological manifestations (e.g., parkinsonism, hearing loss, and/or history of stroke), peripheral neurological manifestations (e.g., limb pain and acroparesthesia), or family history of FD. In individuals with a family history of FD, some of their relatives within the third degree of kinship had already been diagnosed with FD. Individuals who screened positive according to α-Gal A activity in DBS underwent *GLA* sequencing for diagnosis. Twenty-eight individuals (0.4%, 28/8823) with renal manifestations, 29 (0.9%, 29/4057) with cardiac manifestations, and five (0.2%, 5/3075) with central neurological manifestations were diagnosed with FD. FD was mostly detected in individuals with peripheral neurological manifestations (4.4%, 30/894), which are considered important risk factors for early detection of FD. Among other risk-factor groups, FD was diagnosed in 128 of 715 (23.4%) individuals with a family history of FD, and in four of 571 (1.7%) individuals screened for other reasons.

The most frequent variant was p.M296I, which was found in five families. In our screening, these probands were all found in individuals with renal manifestations, although some patients also developed cardiac manifestations [40]. This variant is considered a late-onset variant [41]. The next most common variant was p.R112C, which was found in four families and associated with cardiac symptoms, is considered a classical variant [42]. Seven variants were found in three families: p.R220*, p.R227*, p.K240Efs*8, p.S345Rfs*28, and p.T412Sfs*37 are considered classical variants; and, p.R301Q and p.R112H are considered late-onset variants [42,43,44,45]. Therefore, our high-risk screening effectively detected classical and late-onset FD, as well as heterozygous female patients with FD in the Japanese population. 

## 7. High-Risk Screening Programs for FD

High-risk screening for FD has been performed in different areas worldwide. We conducted a review of the literature on high-risk screening for FD in individuals with renal, cardiac, and central neurological manifestations, with studies lacking a description of detected variants excluded. 

### 7.1. High-Risk Screening for FD in Individuals with Renal Manifestations

We reviewed 34 high-risk screening studies for FD in individuals with renal manifestations (Table 1). Twenty-nine studies screened individuals with end-stage renal failure and those receiving dialysis and/or kidney transplants [46,47,48,49,50,51,52,53,54,55,56,57,58,59,60,61,62,63,64,65,66,67,68,69,70,71,72,73]. Four studies screened individuals with chronic kidney disease before end-stage renal failure [74,75,76,77,78], and one study screened individuals with reduced estimated glomerular filtration rate (eGFR) [79]. In this manifestation category, large-scale screening studies were conducted because more subjects met the selection criteria.

In these studies, pathogenic variants in the *GLA* gene were detected at a frequency of up to 1.1% (0–1.8% in males and 0–2.2% in females), with the frequency in half of these studies ranging from 0.1 to 0.3%. Five studies detected no patients with *GLA* pathogenic variants, except for polymorphisms, and two of these five studies screened using Gb3 measurements [63,78], with the remaining three using <600 subjects for screening [72,74,76].

Each large-scale screening study with >5000 subjects [51,52,53] found >10 *GLA* variants. In these studies, the most common variants were p.L415P (classical), p.R363H (late-onset), p.M296I, and p.R112H and found in Argentina [51,68]. Notably, p.M296I has been detected as a late-onset variant in Japan [47,66]. In our screening, we also detected p.M296I in four patients with renal manifestations [13]. p.R112H has previously been detected in Argentina [68], the Czech Republic [61], Austria [50], and Turkey [70] and in two patients with renal manifestations in our high-risk screening [13]. Therefore, p.R363H, p.M296I, and p.R112H might be late-onset variants with a tendency toward the development of renal manifestations.

Most high-risk screening in individuals with renal manifestations targeted patients that had undergone dialysis and/or renal transplantation due to end-stage renal failure. Therefore, the renal manifestations in patients detected by screening could not be improved with ERT or chaperone therapy. High-risk screening studies targeting individuals with renal manifestations prior to end-stage renal failure have also detected FD patients [75,77,79]. Screening at an earlier symptomatic stage (e.g., proteinuria, albuminuria, and decreased eGFR) is considered to contribute to improving clinical outcomes in patients based on them receiving effective early medical treatment. The α-Gal A activity assay using DBS would be effective in investigating a large population of individuals presenting such symptoms. However, screening by measuring α-Gal A activity alone might miss female patients, making screening with *GLA* sequencing essential. Yalın et al. [53] performed *GLA* sequencing to screen 2215 females with hemodialysis or kidney transplant and diagnosed four (0.2%) heterozygous female patients. This frequency did not differ significantly from that obtained by measuring α-Gal A activity (0–0.3%).

### 7.2. High-Risk Screening for FD in Individuals with Cardiac Manifestations

We reviewed 23 high-risk screenings for FD in individuals with cardiac manifestations (Table 2), with 20 screenings performed for individuals with either LVH or HCM [40,80,81,82,83,84,85,86,87,88,89,90,91,92,93,94,95,96,97,98]. The other three screenings were performed in individuals with a variety of cardiac symptoms, including coronary artery disorders, conduction disorders, cardiomyopathy, and valvular disease [99], those with conduction disorders requiring a pacemaker [100], and those with left ventricular noncompaction (LVNC) [101].

The frequency of pathogenic variants in the *GLA* gene detected in these screenings was up to 3.3% (0–3.8% in males and 0–4.2% in females). That of FD was >1% in almost 50% of these screening studies and higher than that observed in screenings of individuals with renal manifestations. Furthermore, five studies had no patients with pathogenic *GLA* variants. These included screening using urinary Gb3 in individuals with LVH [89], serum α-Gal A activity in males with LVH [85], measurement of α-Gal A activity and/or *GLA* sequencing in individuals with conduction disorders requiring a pacemaker [100], measurement of α-Gal A activity and/or *GLA* sequencing in individuals with LVNC [101], and *GLA* sequencing in individuals with LVH [98]. 

The most common variant detected in the screening was IVS4+919G>A in Hong Kong and Japan, with a total of 11 individuals carrying this variant [82,92,97]. This variant was frequently detected in NBS in Taiwan and patients with late-onset FD with cardiac manifestations [17]. Additionally, p.N215S was detected in a total of 11 patients in Italy, the United States, the United Kingdom, France, the Czech Republic, and Australia [80,81,84,87,91,93]. Moreover, this variant has been frequently detected in patients with late-onset FD with cardiac manifestations [102], with one such patient detected in our high-risk screening in Japan [13]. p.R301Q was detected in two individuals in Korea and Turkey [94,96]. Individuals with this variant present a variety of symptoms, even within the same family [45]. In our high-risk screening, this variant was detected in two individuals with renal manifestations [13].

Overall, high-risk screening in individuals with cardiac manifestations more frequently detected *GLA* pathogenic variants than in individuals with renal manifestations. This might be due to the higher frequency of variants, such as IVS4+919G>A and p.N215S, which tend to develop cardiac manifestations. Although most screenings in individuals with LVH or HCM excluded individuals with hypertension and valvular disease, Fan et al. [82] detected IVS4+919G>A in individuals with LVH and hypertension or valvular disease and insisted that manifestations, such as hypertension and valvular disease, should not be excluded. Conversely, screening in individuals with manifestations other than LVH or HCM, such as conduction disturbances and LVNC, did not detect many *GLA* pathogenic variants. Therefore, screening in individuals with cardiac manifestations other than LVH or HCM might not be as efficient as screening for FD.

Most high-risk screening studies on individuals with cardiac manifestations have been performed by measuring α-Gal A activity. Screening using urinary Gb3 detected no patients with FD [89], suggesting that urinary Gb3 levels would not be a suitable screening test for cardiac FD [23]. Elliot et al. [91] screened 501 females with HCM using *GLA* sequencing and detected two *GLA* pathogenic variants with a frequency of 0.4%, which did not differ significantly from that determined by screening females using α-Gal A activity (0–4.2%).

### 7.3. High-Risk Screening for FD in Individuals with Central Neurological Manifestations

We reviewed 19 studies [103,104,105,106,107,108,109,110,111,112,113,114,115,116,117,118,119,120,121] on high-risk screening in individuals with central neurological manifestations (Table 3). Most of these studies reported screening in individuals with idiopathic strokes. A feature of FD-related central neurological manifestations is juvenile stroke; therefore, 16 studies restricted the upper age limit of the target individuals to between 49 and 60 years. The number of screening subjects ranged from 300 to 1000 in ~50% of the screenings.

The highest frequency (3.7%, 2/54) of *GLA* pathogenic variants was detected in screening performed by measuring α-Gal A activity in individuals aged 22 to 55 years and with idiopathic strokes [108]. Moreover, screening by α-Gal A activity and/or *GLA* sequencing of individuals with acute stroke or transient ischemic attack and aged 18 to 60 years demonstrated that three of 108 subjects (2.7%) harbored *GLA* pathogenic variants [111]. However, no *GLA* pathogenic variant was found in 10 screenings of individuals with central neurological manifestations [103,104,105,106,109,110,114,115,118,119]. Furthermore, p.R227Q was the most frequent variant detected in three individuals in Turkey and France [108,117], with this variant considered to be associated with the development of classical FD [42] and previously detected in one individual during the high-risk screening in individuals undergoing dialysis or kidney transplant in Turkey [53]. Additionally, IVS4+919G>A was detected in two Taiwanese individuals [120].

High-risk screenings of individuals with central nervous system manifestations were performed using a combination of α-Gal A activity and *GLA* sequencing; 50% of these screenings detected no patient with *GLA* pathogenic variants. However, there were FD patients presenting only central neurological manifestations as FD manifestations, suggesting that central neurological manifestations are important for high-risk screening of FD, even if they are less frequent than other symptoms.

## 8. High-Risk Screening to Detect Undiagnosed Patients with FD 

In Japan, the number of areas that perform NBS for FD is expanding; however, screening remains limited. Moreover, even with NBS, there is a possibility of false negatives, especially in females. Therefore, it is possible that there are many undiagnosed patients with FD. High-risk screening by *GLA* sequencing remains a challenge in Japan owing to cost and ethical issues. Thus, in the following sections, we discuss which individuals should be subjected to high-risk screening for FD.

### 8.1. High-Risk Screening in Children and Adolescents

As shown in our high-risk screening, 4.4% of individuals with peripheral neurological manifestations (e.g., limb pain and acroparesthesia) were diagnosed with FD in childhood and adolescence [13]. Therefore, during childhood and adolescence, and to detect classical FD patients and some heterozygous females, α-Gal A activity should be measured for individuals with peripheral neurological manifestations, and *GLA* sequencing should be performed in individuals with impaired α-Gal A activity.

Even for females without impaired α-Gal A activity, measurement of α-Gal A activity and/or *GLA* sequencing in male relatives with FD manifestations might lead to a diagnosis. In the absence of such a family history, *GLA* sequencing should be performed if the characteristics of the pain are more Fabry-like (e.g., exacerbated by increased body temperature or exercise) or if other initial FD manifestations are present (e.g., angiokeratoma, cornea verticillata, hypo-anhidrosis, gastrointestinal symptoms, proteinuria, and mulberries in urine).

### 8.2. High-Risk Screening during Adolescence and in Older People 

During adolescence and at old ages, individuals with renal, cardiac, and central neurological manifestations should undergo high-risk screening for FD. Moreover, they should receive screening, diagnosis, and treatment as early as possible after developing initial manifestations in order to benefit from treatment [122,123]. For this purpose, initial manifestations of FD (e.g., proteinuria, albuminuria, and decreased eGFR as renal manifestations and LVH as a cardiac symptom) should be targeted for screening. Many individuals with these initial manifestations of FD could potentially be detected during medical check-ups. α-Gal A activity measurement in DBS is considered an appropriate screening method for individuals with an initial manifestation of FD during a medical check-up, given the ease of use and utility of DBS. 

### 8.3. GLA Sequencing and Genetic Counseling for Families of Patients with FD

In many high-risk screenings for FD, *GLA* sequencing was performed on family members of the proband, and many cases of FD have been diagnosed by family analysis [124].

There are no ethical issues associated with *GLA* sequencing for the diagnosis of family members that have already developed FD manifestations. However, if *GLA* sequencing is performed on family members that have not developed FD manifestations, genetic counseling should be provided to explain the significance of *GLA* sequencing and the potential impact of the results on family members.

One of the most difficult aspects of genetic counseling for FD is that heterozygous females can vary from asymptomatic to developing manifestations as severe as those observed in males with classical FD. Unfortunately, there is no way to predict whether a heterozygous female will develop the disease. A previous study reported that most heterozygous FD females >68 years of age develop LVH [125]. Therefore, it is necessary to follow up with heterozygous females that have not developed FD manifestations, given that they might eventually develop FD manifestations.

## 9. Next-Generation Screening Method

Lyso-Gb3 in the blood is used not only as a biomarker to evaluate the treatment effect [126] but also as a screening method, as described above. However, Lyso-Gb3 levels in the blood may not always be elevated in patients with late-onset FD or heterozygous female patients, making their use as a marker unsuitable for all FD patients. Currently, microRNAs are being studied as biomarkers and may be expected to be effective as new screening methods [127,128]. 

## 10. Conclusions

FD manifestations vary depending on disease type and patient sex and age; therefore, it is important to establish a high-risk screening system that considers the age of the target population. The peripheral neurological manifestations observed in children and adolescents are specific to FD, making the probability of FD in these cases high. The renal, cardiac, and central neurological manifestations observed during adolescence and in older people are not specific for FD, and the frequency of FD is low, even among those with a high risk of FD. However, it is essential to establish and sustain a system capable of definitively diagnosing FD for physicians (especially pediatricians, cardiologists, nephrologists, and neurologists) in order to allow early diagnosis of FD.

## Figures and Tables

**Table 1 diagnostics-11-01779-t001:** Previous high-risk screening for FD in individuals with renal manifestations.

Criteria [Patients Screened (M|F), *n*]	Age Range	Primary Screening Test	Patients with Variants (M|F), *n* [Prevalence (M|F), %]†	Detected Variants (*n*)	Reference, Year, Country
Dialysis patients [722 (440|282)]	unknown	Plasma α-Gal A	2 (2|0) [0.3 (0.5|0)]	p.Q357* (1)	Utsumi [46], 2000, Japan
Dialysis patients [514 (514|-)]	20–90	Plasma α-Gal A	5 (5|-) [1.0 (1.0|-)]	p.M296I (3), p.A97V (1), p.G373D (1), p.E66Q (1)#	Nakao [47], 2003, Japan
Dialysis patients [2480 (1516|964)]	61.8 (mean)	DBS α-Gal A	4 (4|0) [0.2 (0.3|0)]	p.A121P(1), p.W162R (1), p.I239T (1), p.R112H (1)	Kotanko [50], 2004, Austria
Dialysis patients [450 (450|-)]	26–89	Plasma α-Gal A	1 (1|-) [0.2 (0.2|-)]	g.10252_10254del3 (1)	Ichinose [60], 2005, Japan
Dialysis patients [3370 (1521|1849)]	unknown	DBS α-Gal A	4 (3|1) [0.1 (0.2|0.1)]	p.G360S (1), p.R112H (1), p.I317T (1), p.Q280K (1), p.A143T (1)#	Merta [61], 2007, Czech Republic
Dialysis patients [922 (180|742)]	18–80 (males), >16 (females)	DBS α-Gal A	1 (1|0) [0.1 (0.6|0)]	p.W236R (1), p.A143T (2)#	Terryn [62], 2007, Belgium
CKD patients involving dialysis and transplant patients [499 (499|-)]	63 (mean)	Plasma α-Gal A	0 (0|-) [0 (0|-)]		Andrade [74], 2008, Canada
Dialysis patients [480 (311|169)]	>40 (male), >50 (female)	Serum Gb3	0 (0|0) [0 (0|0)]		Kim [63], 2010, Korea
Dialysis patients [911 (543|368)]	20–91	DBS α-Gal A	2 (1|1) [0.2 (0.2|0.3)]	c.595_596insG (1), c.1037delG (1), p.R118C (3)#, p.D313Y (2)#	Gaspar [64], 2010, Spain
Dialysis patients [933 (557|376)]	unknown	DBS α-Gal A	1 (0|1) [0.1 (0|0.3)]	p.A73E (1), p.E66Q (2)#	Nishino, et al. [65], 2012, Japan
Dialysis patients [1080 (1080|-)]	63.4 (mean)	Plasma α-Gal A	2 (2|-) [0.2 (0.2|-)]	p.G195V (1), p.M296I (1), p.E66Q (8)#	Doi [66], 2012, Japan
Dialysis patients [1136 (615|521)]	18–90	DBS α-Gal A	2 (2|0) [0.2 (0.3|0)]	p.L275F (1), p.P214S (1)	Okur [58], 2013, Turkey
Dialysis patients [1453 (1453|-)]	25–95	Plasma α-Gal A, Plasma Lyso-Gb3	1 (1|-) [0.1 (0.1|-)]	p.Y173* (1), p.E66Q (9)#	Maruyama [67], 2013, Japan
Dialysis patients [1401 (1401|-)]	unknown	DBS α-Gal A	2 (2|-) [0.1 (0.1|-)]	p.R363H (1), p.R112H (1)	Rozenfeld [68], 2015, Argentina
CKD stage1-5 [313 (167|146)]	18–70	DBS α-Gal A	3 (3|0) [1.0 (1.8|0)]	p.N34H (1), c.1072_1074del(p.358delE) (1), p.F229V (1)	Turkmen [75], 2016, Turkey
Dialysis patients [8547 (5408|3139)]	5–98	Plasma α-Gal A	2 (2|0) [0.02 (0.04|0)]	p.R112C (2), p.E66Q (11)#	Saito [69], 2016, Japan
Dialysis patients [1527 (847|680)]	60.2 (mean)	DBS α-Gal A	4 (4|0) [0.3 (0.5|0)]	p.M296V (1), p.R112H (1), p.S65N (1), c.1212_1214delAAG (1), p.D313Y (1)#	Sayilar [70], 2016, Turkey
Dialysis patients [2.583 (2583|-)]	18–91	DBS α-Gal A	3 (3|-) [0.1 (0.1|-)]	p.W204* (1), p.A368T (1), p.C52F (1)	Silva [71], 2016, Brazil
Dialysis patients [142 (81|61)]	20–60	DBS α-Gal A	0 (0|0) [0 (0|0)]		Trachoo [72], 2017, Thailand
CKD not on dialysis [1453 (783|656)]	59.3 (mean)	DBS α-Gal A	0 (0|0) [0 (0|0)]	p.D313Y (2)#, p.A143T (1)#	Yeniçerioğlu [76], 2017, Turkey
Kidney transplant recipients [1095 (648|447)]	unknown	DBS α-Gal A (male), *GLA* sequencing (female)	1 (1|0) [0.1 (0.2|0)]	p.Q330R (1), p.D313Y (2)#, p.S126G (1)#	Yılmaz [73], 2017, Turkey
Dialysis patients [108 (108|-)]	unknown	DBS α-Gal A	1 (1|-) [0.9 (0.9|-)]	p.G35V (1)	Veloso [48], 2018, Brazil
Dialysis patients [227 (148|79)]	65 (mean)	*GLA* sequencing	1 (1|0) [0.4 (0.7|0)]	p.I91T (1), p.D313Y (1)#	Zizzo [49], 2018, Italy
Pre-end stage renal disease [1012 (1012|-)]	20–85	DBS α-Gal A	6 (6|-) [0.6 (0.6|-)]	p.T410A (1), p.G318E (1), IVS4+919G>A (3), p.P210S	Lin [77], 2018, Taiwan
Dialysis patients [9604 (9604|-)]	18–100	DBS α-Gal A	24 (24|-) [0.2 (0.2|-)]	p.C56S (1), p.L415P (4), c.640-1G>C (1), p.R363H (3), p.G85D (1), c.886_887delAT (1), Deletion exons 3-4 (2), c.1235_1236delCT(p.T412fs) (1), p.D109G (2), p.W81* (1), p.P205S (1), p.P409S (1), p.A156_A160del (1), c.902_905insTGTC (1), p.D55G (1), p.G80D (1)	Frabasil [51], 2019, Argentina
Dialysis patients [5572 (3551|2021)]	18–59	DBS α-Gal A	20 (19|1) [0.4 (0.5|0.05)]	p.F273S (1), p.L54P (3), p.N215S (1), p.Y134D (1), p.W262* (1), p.R220* (2), p.R342Q (1), p.D170Y (1), p.W399* (1), p.R227* (1), p.W204C (1), p.E7* (1), p.G183S (1), p.A37S (1), p.C56S (1), p.L68P (1), p.G328R (1)	Moiseev [52], 2019, Russia
Dialysis patients and patients with transplant [5657 (3442| 2215) ]	48 (mean)	DBS α-Gal A (male), *GLA* sequencing (female)	13 (9|4) [0.2 (0.3|0.2)]	p.R227Q (1), p.Q330R (1), p.V199M (1), p.P205T (1), p.E59V (1), IVS6-10G>A (1), p.L54F (1), p.358delE (1), p.S364C (1), p.T39S (1), p.L8Q (1), p.P205S (1), p.C223* (1), p.S126G (4)#, p.D313Y (19)#	Yalın [53], 2019, Turkey
Dialysis patients [526 (325|201)]	unknown	DBS α-Gal A	0 (0|0) [0 (0|0)]		Jahan [54], 2020, Australia
Kidney transplant recipients [265 (175|90)]	53.6 (mean)	*GLA* sequencing, DBS α-Gal A, DBS Lyso- Gb3	3 (1|2) [1.1 (0.6|2.2)]	p.F113L (1), p.D615H (1), p.R220* (1), p.D313Y (2)#, p.S126G (2)#	Veroux [55], 2020, Italy
Kidney transplant recipients [301 (180|120)]	43 (mean)	*GLA* sequencing	1 (1|0) [0.3 (0.6|0)]	c.1093_1101dup (1), p.D313Y (1)#, p.A143T (1)#	Erdogmus [56], 2020, Turkey
CKD stage3-5 [397 (279|118)]	32–75	Dried urine spots Gb3	0 (0|0) [0 (0|0)]		Auray-Blais [78], 2020, Canada
Dialysis patients [619 (unknown)]	>18	DBS α-Gal A	3 (0|3) [0.5 (unknown)]	p.A352G (3)	Alhemyadi [57], 2020, Saudi Arabia
Reduced estimated glomerular filtration rate [1084 (505|579)]	20–70	DBS α-Gal A (male), DBS α-Gal A and *GLA* sequencing (female)	1 (0|1) [0.1 (0|0.2)]	p.L300F (1)	Reynolds [79], 2021, UK
Dialysis patients and patients with transplant [819 (819|-)]	18–70	DBS α-Gal A	1 (1|-) [0.1 (0.1|-)]	p.F113L (1)	Vigneau [59], 2021, France

CKD: Chronic kidney disease, DBS: Dried blood spot; †: Excluding polymorphisms; #: Polymorphisms (p.E66Q, p.D313Y, p.A143T, p.S126G and p.R118C). Mutation nomenclature followed the guidelines established by the Human Genome Variation Society (http://varnomen.hgvs.org/, accessed on 10 September 2021).

**Table 2 diagnostics-11-01779-t002:** Previous high-risk screening for FD in individuals with cardiac manifestations.

Criteria [Patients Screened (M|F), *n*]	Age Range	Primary Screening Test	Patients with Variants (M|F), *n* [Prevalence (M|F), %]†	Detected Variants (*n*)	Reference, Year, Country
LVH [230 (230|-)]	16–87	Plasma α-Gal A	7 (7|-) [3.0 (3.0|-)]	p.M296I (1), p.A20P (1)	Nakao [40], 1995, Japan
HCM [153 (153|-)]	8–71	Plasma α-Gal A	5 (5|-) [3.3 (3.3|-)]	p.N215S (3), p.I317T (1), c.1223del (1), p.D313Y (1)#	Sachdev [80], 2002, UK
HCM [508 (328|108)]	58 (mean)	Plasma α-Gal A	4 (3, 1) [0.8 (0.9|0.9)]	p.L89P (1), p.E358del (1), p.S238N (2), p.A143T (1)#	Monserrat [83], 2007, Spain
HCM [392 (278|114)]	18–79	DBS α-Gal A	4 (4, -) [1.0 (1.4|0)]	p.T162C (1), p.F113L (2), p.N215S (1)	Hagege [84], 2011, France
HCM [1386 (885|501)]	58 (mean)	*GLA* sequencing	4 (2|2) [0.3 (0.2|0.4)]	p.N215S (2), p.D244N (1), p.T410A (1), p.R118C (1)#, p.A143T (2)#	Elliott [91], 2011, 13 European countries
LVH [738 (738|-)]	unknown	Serum α-Gal A	0 (0|-) [0 (0|-)]	p.E66Q (3)#	Mawatari [85], 2013, Japan
LVH [540 (362|178)]	19–93	DBS α-Gal A (male), *GLA* sequencing (female)	1 (1|0) [0.2 (0.3|0)]	p.A5E (1), p.A143T (4)#	Terryn [86], 2013, Belgium
LVH [100 (100|-)]	33–83	DBS α-Gal A	4 (4|-) [4.0 (4.0|-)]	intronic splice variants (2), p.N215S (1), p.Y152H (1)	Palecek [87], 2014, Czech Republic
LVH [47 (23|24)]	25–90	DBS α-Gal A	1 (0|1) [2.1 (0|4.2)]	p.W262L (1)	Baptista [88], 2015, Portugal
LVH [2596 (1689|907)]	64 (mean)	Urinary Gb3	0 (0|0) [0 (0|0)]		Gaggl [89], 2016, Austria
HCM [273 (169|104)]	58 (mean)	DBS α-Gal A	3 (1|2) [1.1 (0.6|1.9)]	p.W226* (1), p.N224S (1), c.547+3A>G (1)	Seo [90], 2016, Korea
HCM [177 (unknown)]	15–87	Plasma α-Gal A	2 (unknown) [1.1 (unknown)]	p.R112L (1), IVS4+919G>A (1)	Kubo [92], 2017, Japan
Coronary artery disease, conduction or rhythm abnormalities, non-ishemic cardiomyopathy, and valvular dysfunction [2256 (1404|852)]	19–95	Urinary Gb3, DBS α-Gal A, *GLA* sequencing	1 (0|1) [0 (0|0.1)]	p.D83N (1), p.D313Y (7)#, p.R118C (2)#	Schiffmann [99], 2018, USA
HCM [585 (413|172)]	50 (mean: male), 57 (mean: female)	DBS α-Gal A (male), *GLA* sequencing (female)	2 (1|1) [0.3 (0.2|0.6)]	p.N215S (2)	Maron [93], 2018, USA
LVH [986 (986|-)]	unknown	Plasma α-Gal A	3 (3|-) [0.3 (0.3|-)]	p.G328R (1), p.R301Q (1), p.H46R (1), p.E66Q (2)#	Kim [94], 2019, Korea
LVH [277 (215|62)]	25–79	Plasma α-Gal A, Plasma Lyso-Gb3	2 (1|1) [0.7 (0.5|1.6)]	intronic splice variants (1), p.R112H (1), p.E66Q (2)#	Yamashita [95], 2019, Japan
Conduction disordaers requiring a pacemaker [188 (124|64)]	63 (mean)	α-Gal A (male), *GLA* sequencing (female)	0 (0|0) [0 (0|0)]	p.D313Y (1)#	Lopez-Sainz [100], 2019, Spain
HCM [80 (53|27)]	18–65	*GLA* sequencing	2 (2|0) [2.5 (3.8|0)]	p.R112C (1), p.R301Q (1)	Barman [96], 2019, Turkey
LVNC [78 (49|29)]	47 (mean)	DBS α-Gal A (male), *GLA* sequencing (female)	0 (0|0) [0 (0|0)]		Azevedo [101], 2019, Portugal
LVH [190 (119|71)]	47.2 (mean)	*GLA* sequencing	0 (0|0) [0 (0|0)]	p.A143T (1)#, p.D313Y (1)#	Barman [98], 2019, Turkey
LVH [266 (167|99)]	27–98	DBS α-Gal A	5 (5|0) [1.9 (3.0|0)]	IVS4+919G>A (5)	Sadasivan [97], 2020, Canada and Hong Kong
LVH [511 (332|179)]	18–75	DBS α-Gal A	1 (1|0) [0.2 (0.3|0)]	p.N215S (1), p.A143T (1)#	Fuller [81], 2020, Australia
LVH [499 (336|163)]	66.4 (mean)	DBS α-Gal A	8 (8|0) [1.6 (2.4|0)]	IVS4+919G>A (8)	Fan [82], 2021, Hong Kong

LVH: Left ventricular hypertrophy, HCM: Hypertrophic cardiomyopathy, DBS: Dried blood spot, LVNC: left ventricular noncompaction; †: Excluding polymorphisms; #: Polymorphisms (p.E66Q, p.D313Y, p.A143T, p.S126G and p.R118C). Mutation nomenclature followed the guidelines established by the Human Genome Variation Society (http://varnomen.hgvs.org/, accessed on 10 September 2021).

**Table 3 diagnostics-11-01779-t003:** Previous high-risk screening for FD in individuals with central neurological manifestations.

Criteria [Patients Screened (M|F), *n*]	Age Range	Primary Screening Test	Patients with Variants (M|F), *n* [Prevalence (M|F), %]†	Detected Variants (*n*)	Reference, Year, Country
Cryptogenic stroke [103 (64|39)]	16–60	DBS α-Gal A	0 (0|0) [0 (0|0)]		Brouns [103], 2007, Belgium
First ischemic stroke [558 (558|-)]	15–49	Plasma α-Gal A	0 (0|-) [0 (0|-)]	p.143T (1)#, p.D313Y (1)#	Wozniak [104], 2010, USA
Stroke, unexplained white matter lesions, or vertebrobasilar dolichoectasia [1000 (547|453)]	18–60	DBS α-Gal A (male), *GLA* sequencing (female)	0 (0|0) [0 (0|0)]	p.A143T (2)#, p.S126G (1)#, p.D313Y (5)#	Brouns [110], 2010, Belgium
First stroke [493 (300|193)]	18–55	*GLA* sequencing	0 (0|0) [0 (0|0)]	p.R118C (6)#, p.D313Y (6)#	Baptista [114], 2010, Portugal
Stroke or transient ischemic attack [1046 (502|544)]	24–103	*GLA* sequencing	0 (0|0) [0 (0|0)]	p.D313Y (5)#	Marquardt [115], 2012, UK
Stroke or transient ischemic attack [150 (102|48)]	18–55	Serum α-Gal A	0 (0|0) [0 (0|0)]		Sarikaya [106], 2012, Switzerland
Stroke or transient ischemic attack [5023 (2962|2061)]	18–55	*GLA* sequencing	unknown [unknown]	p.V315I (1), p.D83N (1), p.L415F (1), p.S102L (1), p.E418G (1),p.R118C (1)#, p.S126G (3)#, p.A143T (4)#, p.D313Y (4)#	Rolfs [116], 2013, 15 European countries
Cryptogenic ischemic stroke [100 (55|45)]	16–55	*GLA* sequencing, Plasma Lyso-Gb3	1 (1|0) [1.0 (1.8|0)]	intronic splice variants (1)	Dubuc [121], 2013, Canada
Stroke, transient ischemic attack, white matter lesions, or cerebral venous thrombosis [178 (73|105)]	18–55	*GLA* sequencing	1 (1|0) [0.6 (1.4|0)]	p.R227Q (1), p.D313Y (1)#	Fancellu [117], 2015, Italy
Lacunar stroke [994 (706|288)]	56.7 (mean)	*GLA* sequencing	0 (0|0) [0 (0|0)]	p.R118C (1)#	Kilarski [118], 2015, UK
Acute ischemic stroke or transient ischemic attack [108 (66|42)]	18–60	Leukocyte α-Gal A (male), *GLA* sequencing (female)	3 (1|2) [2.7 (1.5|4.8)]	p.R301G (1), p.L415R (1), c.548-3_553del (1)	Romani [111], 2015, Italy
Stroke [588 (363|225)]	74.1 (mean)	DBS α-Gal A	1 (0|1) [0.2 (0|0.4)]	p.M1T (1), p.E66Q (7)#	Nagamatsu [107], 2017
Cryptogenic stroke [54 (30|24)]	24–55	DBS α-Gal A	2 (2|0) [3.7 (6.7|0)]	p.R227Q (2)	Gündoğdu [108], 2017, Yurkey
Cryptogenic ischemic stroke or transient ischemic attack [397 (218|179)]	18–55	*GLA* sequencing	0 (0|0) [0 (0|0)]	p.R118C (1)#, p.D313Y (1)#	Lanthier [119], 2017, Canada
Stroke or transient ischemic attack [311 (168|143)]	18–55	DBS α-Gal A (male), *GLA* sequencing (female)	1 (0|1) [0.3 (0|0.7)]	p.M296I (1), p.S126G (1)#, p.D313Y (2)#	Reisin [112], 2018, Argentina
Ischemic stroke or intracerebral hemorrhage [516 (396|120)]	21–60	Plasma α-Gal A, Plasma Lyso-Gb3	0 (0|0) [0 (0|0)]	p.E66Q (2)#	Kinoshita [105], 2018, Japan
Ischemic stroke or intracerebral hemorrhage [1000 (750|250)]	18–55	*GLA* sequencing (26 variants)	2 (2|0) [0.2 (0.3|0)]	IVS4+919G>A (2)	Lee [120], 2019, Taiwan
Cryotogenic stroke [50 (27|23)]	48–56	DBS α-Gal A	0 (0|0) [0 (0|0)]	p.R118C (1)#	Malavera [109], 2020, Australia
Cryptogenic stroke [114 (75|39)]	18–50	DBS α-Gal A (male), *GLA* sequencing (female)	3 (2|1) [2.6 (2.7|2.6)]	p. F113I (1), intronic splice variants (2)	Afanasiev [113], 2020, Israel

DBS: Dried blood spot; †: Excluding polymorphisms; #: Polymorphisms (p.E66Q, p.D313Y, p.A143T, p.S126G and p.R118C). Mutation nomenclature followed the guidelines established by the Human Genome Variation Society (http://varnomen.hgvs.org/, accessed on 10 September 2021).

## Data Availability

Not applicable.

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
