# Peer review of "High-Risk Screening for Fabry Disease: A Nationwide Study in Japan and Literature Review"

_diagnostics, 2021, doi:10.3390/diagnostics11101779_

Round 1

Reviewer 1 Report

This is a report on a Fabry disease screening study with a comprehensive review on earlier screening studies. While many similar screening studies have been published from several countries and populations, the review itself appears worth publishing. I have only minor issues.

- While the screening studies are nicely listed, here is no i.e. Table nicely summarizing the findings, to put the many seperate studies in a larger context

  • Introduction: "angiohemangioma" - pls check
  • Methods: "However, depending on the region of the GLA gene to be analyzed, there are some gene variants that cannot be detected. Intronic variants, such as IVS4+919G>A, are not detected during the analysis of exons and/or in-tron regions close to exons. Conversely, ..." any literature available?
  • Methods: "In our high-risk screening algorithm, individuals showing low DBS α-Gal A activity (< a cut-off ) in the two tests underwent GLA gene sequencing.“ - Please explain further why and how two tests were used.
  • High-risk screening: Part of the study population aprears to be high risk screeing based on "family history". Please explain further. 
  • “High-risk screenings of individuals with central nervous system manifestations were performed using a combination of α-Gal A activity and GLA sequencing; however, 50 % of these screenings detected no patient with GLA pathogenic variants. Because there were patients presenting only central neurological manifestations as FD manifestations, this suggests that central neurological manifestations are important for high-risk screening of FD.“ Pls check - one might argue that due to low prevalence this might rather be called "low-risk" screening.

  • "Future high-risk screening
    • Gene therapy and substrate-reducing therapy for FD are currently undergoing clini-cal trials [118]. Approval of these therapies will increase treatment options but also intro-duce new problems regarding which treatments should be given." Meaning of this sentence in th given context is questionable, pls check.

Author Response

Reviewer 1

This is a report on a Fabry disease screening study with a comprehensive review on earlier screening studies. While many similar screening studies have been published from several countries and populations, the review itself appears worth publishing. I have only minor issues.

While the screening studies are nicely listed, here is no i.e. Table nicely summarizing the findings, to put the many seperate studies in a larger context

Point 1

Introduction: "angiohemangioma" - pls check

Response: Thank you for your pointing this out. The term has been changed to "angiokeratoma".

Point 2

Methods: "However, depending on the region of the GLA gene to be analyzed, there are some gene variants that cannot be detected. Intronic variants, such as IVS4+919G>A, are not detected during the analysis of exons and/or in-tron regions close to exons. Conversely, ..." any literature available?

Response: Thank you for your comment. The following literature about IVS4+919G>A has been added to the relevant text in the revised manuscript.

Ishii, S.; Nakao, S.; Minamikawa-Tachino, R.; Desnick, R.J.; Fan, J.Q.

Alternative splicing in the α-galactosidase A gene: Increased exon inclusion results in the

Fabry cardiac phenotype. Am. J. Hum. Genet. 2002, 70, 994–1002.

Point 3

Methods: "In our high-risk screening algorithm, individuals showing low DBS α-Gal A activity (< a cut-off ) in the two tests underwent GLA gene sequencing.“ - Please explain further why and how two tests were used.

Response: Thank you for your important comment. The related sentence has been modified as follows;

“ In our high-risk screening algorithm, individuals showing low DBS α-Gal A activity (< a cut-off) in the two times tests (first and second assays) underwent GLA gene sequencing”. (Page 3, Line 11)

Point 4

High-risk screening: Part of the study population appears to be high risk screening based on "family history". Please explain further. 

Response: Thank you for your comment. To improve clarity, the related sentence has been revised as follows;

“In individuals with a family history of FD, some of their relatives within the third degree of kinship had already been diagnosed with FD.” (Page 3)

Point 5

“High-risk screenings of individuals with central nervous system manifestations were performed using a combination of α-Gal A activity and GLA sequencing; however, 50 % of these screenings detected no patient with GLA pathogenic variants. Because there were patients presenting only central neurological manifestations as FD manifestations, this suggests that central neurological manifestations are important for high-risk screening of FD.“ Pls check - one might argue that due to low prevalence this might rather be called "low-risk" screening.

Response: Thank you for your valuable comment. We have rechecked this portion as suggested; however, because central nervous system symptoms such as juvenile stroke are important risk factors for FD, we choose to retain the original content.

Point 6

"Future high-risk screening

Gene therapy and substrate-reducing therapy for FD are currently undergoing clini-cal trials [118]. Approval of these therapies will increase treatment options but also intro-duce new problems regarding which treatments should be given." Meaning of this sentence in th given context is questionable, pls check.

Response: Thank you for your comment. The description regarding future high-risk screening has been revised and moved to the Introduction section.

Reviewer 2 Report

The manuscript entitled “High-risk Screening for Fabry Disease: A Nationwide Study in Japan and Literature Review” by Sawada et al, is a review concerning with Fabry disease and screening, mainly high-risk screening in patients. The data collected from the literature are quite clearly presented. The authors grouped and analysed the published studies, according to the clinical manifestations (renal, cardiac or central neurological manifestations).

I think the manuscript could be published although I would strongly recommend some modifications.

Here there are my major concerns.

  1. In the ‘Abstract’, the authors state: ‘Here, we performed high-risk screening for FD in Japan and showed….’. Do the authors mean that in this paper they describe for the first time these results?

It seems to me that they summarize a previous study they conducted (reference 5) in the paragraph 6 “ High-risk screening in Japan”. Are you going to show in this paper new, unpublished results? So please clarify it and modify the text accordingly.

  1. In the ‘Introduction’ the authors only cite the Enzyme-replacement therapy (ERT) as the currently-available treatment option for FD.

Although not yet used for long, a pharmacological chaperone therapy approved by EMA and by FDA, is currently available. The therapeutic approach based on the so called pharmacological chaperones is nowadays quite largely used to treat several diseases (http://dx.doi.org/10.3390/molecules25143145; http://dx.doi.org/10.3390/ijms21020489). The pharmacological chaperone available for Fabry disease is 1-deoxy-galactonojirimycin, DGJ, commercial name Galafold™, (https://doi.org/10.1002/jimd.12228, https://doi.org/10.1007/s40265-019-01090-4 ). A pharmacological chaperone specifically binds to the target enzyme and stabilize it (https://doi.org/10.1186/jbiol186 ).  DGJ is an inhibitor of the lysosomal a-galctosidase, and as a consequence a precise dosage regimen need to be applied, in order to balance the stabilizing effect and the inhibitory one. Moreover, only mutations that affect the stability of the lysosomal a-galctosidase but not its enzymatic activity, are amenable to the treatment with DGJ. This means that the responsiveness of each mutation needs to be carefully considered (https://doi.org/10.1186/1750-1172-8-111, https://doi.org/10.3390/ijms17122010, https://doi.org/10.1016/j.cca.2018.02.021 )

I think these aspects, even if shortly, need to be presented in the introduction section.

  1. Moreover, I do not understand the need to have the paragraph 9 ‘Future high-risk screening’. I suggest to eliminate this paragraph and to move elsewhere its content. It seems to me that the ‘introduction’ is a good place where to give a short and comprehensive overview of the therapeutic approaches.

Minor concerns.

I suggest a horizontal layout for the tables.

Moreover, please check and uniform the style within the tables, specifically the symbol of thousands (comma or point?).  

In table 1, 2 and 3, there are some asterisks whose meanings are not specified. For example ‘p.Q357* (1), p.W204* (1)’ and many others. 

Author Response

Reviewer 2

The manuscript entitled “High-risk Screening for Fabry Disease: A Nationwide Study in Japan and Literature Review” by Sawada et al, is a review concerning with Fabry disease and screening, mainly high-risk screening in patients. The data collected from the literature are quite clearly presented. The authors grouped and analysed the published studies, according to the clinical manifestations (renal, cardiac or central neurological manifestations).

I think the manuscript could be published although I would strongly recommend some modifications.

Here there are my major concerns.

Point 1

In the ‘Abstract’, the authors state: ‘Here, we performed high-risk screening for FD in Japan and showed’. Do the authors mean that in this paper they describe for the first time these results?

It seems to me that they summarize a previous study they conducted (reference 5) in the paragraph 6 “ High-risk screening in Japan”. Are you going to show in this paper new, unpublished results? So please clarify it and modify the text accordingly.

Response: Thank you for your important comment. New results were not included in our study. Therefore, the following sentence has been provided to improve clarity.

“The outcome of the high-risk screening in Japan is summarized below.”

Point 2

In the ‘Introduction’ the authors only cite the Enzyme-replacement therapy (ERT) as the currently-available treatment option for FD.

Although not yet used for long, a pharmacological chaperone therapy approved by EMA and by FDA, is currently available. The therapeutic approach based on the so called pharmacological chaperones is nowadays quite largely used to treat several diseases (http://dx.doi.org/10.3390/molecules25143145; http://dx.doi.org/10.3390/ijms21020489). The pharmacological chaperone available for Fabry disease is 1-deoxy-galactonojirimycin, DGJ, commercial name Galafold™, (https://doi.org/10.1002/jimd.12228, https://doi.org/10.1007/s40265-019-01090-4). A pharmacological chaperone specifically binds to the target enzyme and stabilize it (https://doi.org/10.1186/jbiol186 ).  DGJ is an inhibitor of the lysosomal a-galctosidase, and as a consequence a precise dosage regimen need to be applied, in order to balance the stabilizing effect and the inhibitory one. Moreover, only mutations that affect the stability of the lysosomal a-galctosidase but not its enzymatic activity, are amenable to the treatment with DGJ. This means that the responsiveness of each mutation needs to be carefully considered (https://doi.org/10.1186/1750-1172-8-111, https://doi.org/10.3390/ijms17122010, https://doi.org/10.1016/j.cca.2018.02.021 )

I think these aspects, even if shortly, need to be presented in the introduction section.

Response: Thank you for your pertinent suggestion. The description of pharmacological chaperone therapy has been provided as follows in the Introduction section.

“Moreover, a pharmacological chaperone therapy (Migalastat; 1-deoxygalactonojirimycin) was recently approved for FD patients with amenable GLA mutations.”

Point 3

Moreover, I do not understand the need to have the paragraph 9 ‘Future high-risk screening’. I suggest to eliminate this paragraph and to move elsewhere its content. It seems to me that the ‘introduction’ is a good place where to give a short and comprehensive overview of the therapeutic approaches.

Response: Thank you for your advice. “Future high-risk screening” has been revised to “Next-generation high-risk screening,” and the content has been revised and moved to the Introduction section.

.

Minor concerns.

Point 4

I suggest a horizontal layout for the tables.

Moreover, please check and uniform the style within the tables, specifically the symbol of thousands (comma or point?).  

Response: Thank you for your suggestions. The tables have been modified accordingly.

Point 5

In table 1, 2 and 3, there are some asterisks whose meanings are not specified. For example ‘p.Q357* (1), p.W204* (1)’ and many others. 

Response: Thank you for pointing this out. Asterisks indicate “stop codon” according to the guidelines.

Round 2

Reviewer 2 Report

Point 2

In the ‘Introduction’ the authors only cite the Enzyme-replacement therapy (ERT) as the currently-available treatment option for FD.

Although not yet used for long, a pharmacological chaperone therapy approved by EMA and by FDA, is currently available. The therapeutic approach based on the so called pharmacological chaperones is nowadays quite largely used to treat several diseases (http://dx.doi.org/10.3390/molecules25143145; http://dx.doi.org/10.3390/ijms21020489). The pharmacological chaperone available for Fabry disease is 1-deoxy-galactonojirimycin, DGJ, commercial name Galafold™, (https://doi.org/10.1002/jimd.12228, https://doi.org/10.1007/s40265-019-01090-4). A pharmacological chaperone specifically binds to the target enzyme and stabilize it (https://doi.org/10.1186/jbiol186 ).  DGJ is an inhibitor of the lysosomal a-galctosidase, and as a consequence a precise dosage regimen need to be applied, in order to balance the stabilizing effect and the inhibitory one. Moreover, only mutations that affect the stability of the lysosomal a-galctosidase but not its enzymatic activity, are amenable to the treatment with DGJ. This means that the responsiveness of each mutation needs to be carefully considered (https://doi.org/10.1186/1750-1172-8-111, https://doi.org/10.3390/ijms17122010, https://doi.org/10.1016/j.cca.2018.02.021 )

I think these aspects, even if shortly, need to be presented in the introduction section.

Response: Thank you for your pertinent suggestion. The description of pharmacological chaperone therapy has been provided as follows in the Introduction section.

“Moreover, a pharmacological chaperone therapy (Migalastat; 1-deoxygalactonojirimycin) was recently approved for FD patients with amenable GLA mutations.”

I am sorry but the section about pharmacological chaperones is too short. This treatment represents an important option for many patients and should not be overlooked: my word “Even shortly”were misinterpreted and I reformulate the request:

I think these aspects need to be presented in the introduction section.

WITHOUT THE SENTENCE: 

Author Response

Reviewer 2

In the ‘Introduction’ the authors only cite the Enzyme-replacement therapy (ERT) as the currently-available treatment option for FD.

Although not yet used for long, a pharmacological chaperone therapy approved by EMA and by FDA, is currently available. The therapeutic approach based on the so called pharmacological chaperones is nowadays quite largely used to treat several diseases (http://dx.doi.org/10.3390/molecules25143145; http://dx.doi.org/10.3390/ijms21020489). The pharmacological chaperone available for Fabry disease is 1-deoxy-galactonojirimycin, DGJ, commercial name Galafold™, (https://doi.org/10.1002/jimd.12228, https://doi.org/10.1007/s40265-019-01090-4). A pharmacological chaperone specifically binds to the target enzyme and stabilize it (https://doi.org/10.1186/jbiol186 ).  DGJ is an inhibitor of the lysosomal a-galctosidase, and as a consequence a precise dosage regimen need to be applied, in order to balance the stabilizing effect and the inhibitory one. Moreover, only mutations that affect the stability of the lysosomal a-galctosidase but not its enzymatic activity, are amenable to the treatment with DGJ. This means that the responsiveness of each mutation needs to be carefully considered (https://doi.org/10.1186/1750-1172-8-111, https://doi.org/10.3390/ijms17122010, https://doi.org/10.1016/j.cca.2018.02.021 )

I think these aspects, even if shortly, need to be presented in the introduction section.

Response: Thank you for your pertinent suggestion. The description of pharmacological chaperone therapy has been provided as follows in the Introduction section.

“Moreover, a pharmacological chaperone therapy (Migalastat; 1-deoxygalactonojirimycin) was recently approved for FD patients with amenable GLA mutations.”

I am sorry but the section about pharmacological chaperones is too short. This treatment represents an important option for many patients and should not be overlooked: my word “Even shortly”were misinterpreted and I reformulate the request:

I think these aspects need to be presented in the introduction section.

Response: Thank you for your advice. According to your advice, we added the description of a pharmacological chaperone therapy (Migalastat; 1-deoxygalactonojirimycin) again.

“Moreover, a pharmacological chaperone therapy (Migalastat; 1-deoxygalactonojirimycin) was recently approved for FD patients with amenable GLA variants [5]. Migalastat that selectively and reversibly binds to the active sites of amenable mutant forms of α-Gal A enzyme [6,7]. Given its efficacy, convenient oral regimen and the limited therapeutic options available, migalastat can be an important therapeutic option for FD patients with migalastat-amenable GLA variants [8]. Responsiveness of each variant should be carefully evaluated [9–11].”

Round 3

Reviewer 2 Report

the paper can be accepted in the present form